# OpenReview forum: "More of the Same: Persistent Representational Harms Under Increased Representation"
_NeurIPS.cc/2025/Conference — NeurIPS 2025 poster_

### Official Review · Reviewer_keQ9 · 2025-06-15

**Clarity:** 2
**Significance:** 2
**Originality:** 2
**Rating:** 4
**Confidence:** 4

**Summary:**

This paper asks 2 questions related to gender bias in LLMs: (1) has there been a change over time in how LLMs associate gender with occupations (2) do models generate text with gender-associated stereotypes even if the prompt doesn't contain an explicit gender marker. The paper compares one old model (GPT 3.5) with two newer models (GPT 4o mini and Llama 3.1) by prompting the model to generate biographies of given occupations and finds that models have increased in their tendency to generate biographies of women, across occupations. The authors then test whether the texts (generated from prompts without mentioning gender) that describe women/men tend to be more similar to texts describing women/men from prompts with explicit gender, and finds they are, and that this similarity increased from GPT 3.5 to GPT 4omini.

**Questions:**

My questions are mainly highlighted in the weaknesses, but are repeated here for clarity:

1) Why is it unexpected that models that generate biographies of women should use women-associated terms, even if the prompt doesn't mention a woman?

2) For each method, how is it better than prior methods:
a) Calibrated Marked Words
b) SRBS including Chamfer Distance


3) What exactly is the historical claim about  generation: that models generate more biographies of women, or that models generate more association of women with different occupations. Can you distinguish these?

4) What exactly is the historical claim about the relationship of occupations to BLS stats, and are your results for GPT3.5 consistent with earlier papers using GPT3.5 that have suggested underrepresentations of women?

5) How does the Gender Asssociation method differ from prior methods?

6) which of your findings hold with statistical significance and which do not?

7) How are the sterotypes described in section 5.3 different than similar stereotypes already described in the literature?

**Ethical Concerns:**

["NO or VERY MINOR ethics concerns only"]

**Final Justification:**

Paper addresses an important problem, and worries about the unclear relationship with the literature slash  lack of careful description of the innovation of methods have been mitigated by the proposed changes highlighted in the rebuttal.

**Limitations:**

yes

**Paper Formatting Concerns:**

equations are poorly formatted.

**Quality:**

2

**Strengths And Weaknesses:**

Strengths:

(1) Knowing whether models have changed over time in gender stereotypes in their output is an important question

(2) The paper looks at two ways to answer this question, both at the change in the percentage of biographies generated for different genders for different occupations, and for the actual stereotypes generated.

(3) The finding that OpenAI models have increased in their tendency to generate biographies of women (at least between 3.5 and 4o-mini)  is an interesting finding.

Weaknesses:

The main weakness is that overall the method and findings are insufficiently novel and insufficiently well-motivated. Some examples:

(1) One of the two main findings, that biographies about women contain more women-associated terms (even if the prompt doesn't explicitly ask for a biography of a particular gender) is quite unsurprising.  Dozens of papers have shown that these models generate biased gender associations, and many of them use prompts that don't have gender in the prompt (e.g., recent papers like https://aclanthology.org/2025.findings-naacl.398/ or https://doi.org/10.1007/s10579-024-09780-6).  But more worrisome, given that these models have amassed so much evidence that models generate these biased associations, it's difficult to defend the possibility that model might write a biography that describes a woman but fail to generate gender-associated language simply because the prompt didn't mention the gender.

(2) Methods are insufficiently motivated and insufficiently compared to alternative methods.  The calibrated marked words method is not motivated (why remove frequent words? the Fightin' Words method specifically argues the method doesn't need frequent word removal). If this was necessary because of a failing of Fightin' Words to handle this use case, that should be discussed and motivated. If Calibrated Marked Words is to be a contribution, it has to be carefully compared to alternative methods (perhaps even just simple raw log-likelihood). The Gender Association Method (count pronouns and honorifics) is simplistic and not different from what other papers do in this situation (e.g. https://aclanthology.org/2025.findings-naacl.398/ or https://doi.org/10.1007/s10579-024-09780-6).  The Subset Representational Bias score  and the use of Chamfer distance is not motivated or compared to other cosine-based methods for quantifying gender bias, like GenBit (Sengupta,  B.;   Maher,  R.;   Groves,  D.;   Olieman,  C.  (2021).  GenBiT:  measure and  mitigate  gender  bias  in  language  datasets, Microsoft  Journal  of  Applied  Research,   16,   63–71,   2021.)

(3) The finding that there is a historical shift in the percentage of women biographies generated across occupations is interesting but both unclear and insufficiently supported. The main paper simply claims it based on visual evidence in Figure 1. But without any statistical tests, it's hard to know how robust this is. This claim should be tested statistically. In addition, it's hard to know what the exact claim is about this change. Is the claim that GPT4o is generally more likely to generate biographies of women, irrespective of occupation or other attributes?  Or that specifically there is a change in something about the occupation-gender association? These are not distinguished. The reader also can't tell if the claim is simple that models generate more women biographies than BLS, or that there is a historical change and prior models did not generate more women biographies and current models do. (the relationship between GPT3.5 and BLS is not clear in the paper or the appendix). All of the historical claims mainly rest on changes from a single model GPT3.5 to another single model, GPT4o-mini, since differences between GPT3.5 and Llama may be due to architectures or randomly different training data rather than explicit changes in gender associations over time.

(4) The paper is generally weak on statistical tests of any kind, making many of the comparisons feel anecdotal

(5) The stereotypes in seciton 5.3 seem quite similar to stereotypes already described in the very extensive gender bias literature. Are some of these new? if so which ones and what do they contribute to the literature?

(5) The paper is poorly written and poorly checked.  Many terms (associated gender, specified gender) are used well before they are defined.  The equations are not numbered, making it difficult to point out that first equation is wrong ( "min dx(A,B)" should be  "min dx(v,u)") and the second equation has a typo (a period inside the second Chamfer distance). The latex for equations should be cleaned up (just as one example CH is currently not be typeset as two variables C and H).

---

> ### Author Rebuttal · Authors · 2025-07-31
>
> We thank the reviewer for their incredibly detailed review and insight provided! We appreciate all of the questions and feedback provided!!
>
> Our paper’s main contributions are the development of an evaluation methodology to surface representational differences in generated text between groups without specifying group membership in the prompt, and an empirical analysis utilizing this evaluation methodology in the gender and occupation domain, where we demonstrate that gendered associations persist in biographies and personas generated by prompts not specifying gender and these associations correspond to stereotypes and harms outlined in the social science literature. Our evaluation methodology is as follows: 1) generate text, 2) associate each generation with a group, 3) statistically test whether differences between groups are significant, 4) analyze meaningful words for patterns and potential harms, grounded in social science scholarship. To more clearly communicate this, we will add a figure outlining this evaluation methodology and add the following to the beginning of our methodology section:
> > We develop an evaluation methodology to surface representational differences in how groups are represented in generated text displayed in Figure 1. First, we generate text both with and without specifying a group in the prompt. Second, we associate each generation with a group. Third, we statistically test whether differences between groups are significant. We then analyze these meaningful differences for patterns and potential harms grounded in social science scholarship.
>
> >Q1
>
> We agree with the reviewer that it may be expected that women-associated words persist in biographies about women. However, our goal is not to simply show such associations persist. Instead, our **empirical contribution** is to identify statistically significant words that differentiate how men and women are represented in specific occupational contexts—where the relevant associations are not obvious and often differ from generic gender markers (e.g., ‘women’-‘mother’ association). For example, ‘ambitions,’ ‘accomplishments,’ and ‘countless’ are statistically significant words for female software engineers, but only ‘countless’ is statistically significant in both generations where gender is specified and associated. Similarly, the word ‘passion’ is statistically significant for women in many occupations but not broadly in other domains, suggesting occupation-specific representational differences. These nuanced representational differences are essential for understanding how biases persist and proliferate in downstream tasks.
>
> We also emphasize our methodological contribution: an evaluation framework that identifies representational differences between groups in a context-dependent manner. This enables researchers and practitioners to perform a more nuanced, statistically grounded analysis of how different groups are characterized. As LLMs are increasingly used across domains, our method offers a practical tool for identifying and measuring representational biases in generated text.
>
> Finally, the papers cited by the reviewer are important but differ in scope. While both examine gender bias in unprompted contexts, neither investigates how different groups are represented. One paper shows that male bias persists in generated medical cases but does not explore representational differences between men and women or offer a method to analyze them. The other examines gender bias in cover letters across French and Italian but focuses primarily on the frequency of gender markers, without assessing broader patterns in how men and women are described. Moreover, its approach may not generalize well to languages with fewer explicit gender markers. In contrast, our work directly analyzes representational differences and introduces a generalizable methodology for surfacing these patterns—addressing a critical gap in the literature.
> >Q2
>
> The Calibrated Marked Words method is an instance of the Fightin’ Words method in Monroe et al., using a prior that is a mixture of in-distribution context-specific text (i.e., language model-generated biographies), and a corpus representative of English language (see Section 3.1 and A.2 for details). We found this prior was more effective at not including common English words from our list of words with statistically significant differences in usage, while negligibly affecting other (context-specific) statistically significant words, compared to the prior used in the Marked Words method of Cheng et al (which only used in-distribution text as the prior). Other than selecting the hyperparameter governing mixture weights, there is no additional manual removal of common words from our significant words list, and we directly follow the Fightin’ Words methodology.
>
>
> As described in Section 3.3, Chamfer-based SRBS solves the following problem: given a list of words C, and two lists of words T1 and T2, measure how much more similar C is to T1 than to T2. We note that in our setting, all of C, T1, and T2 have already been selected as statistically significant in their respective contexts (i.e., C is from generated personas with inferred genders, T1 is specified male, and T2 is specified female). We use the Chamfer distance to account for simple invariances in the word lists, e.g., synonyms should not be considered different.
>
> We would like to emphasize that other methods in the literature address fundamentally different problems, and are not suited for our task. For instance, the GenBit method solves the following problem: given two lists of words T1 and T2, and a co-occurence matrix, assign each word a score based on how much more it co-occurs with T1 than T2. Importantly, in our setting, T1 and T2 arise from different distributions than C (i.e., use different prompts), so a co-occurrence matrix between the two lists is not meaningful, as it would cross significant words / their meanings from two different contexts. Moreover, GenBit does not appear to encode semantic invariances such as synonyms, beyond basic lemmatization.
>
> Reviewer 2KRr mentioned the WEAT method, but it does not address our goal of comparing differences in similarities between lists. WEAT adapts the Implicit Association Test to evaluate bias in word embeddings and requires carefully curated word sets to be meaningful (e.g., male/female names vs. career/family terms). WEAT relies on the mean cosine distance. The choice of mean of cosine distance is arguably appropriate for WEAT because the lists of attributes are carefully curated such that each list of attributes refers to a particular category (e.g. family). While we could use mean cosine distance (and doing so would not affect the novelty of our proposed contribution), we believe Chamfer distance is much preferable because the interpretation and value of the mean cosine distance is less clear when the lists are not curated and may differ from one another in a multitude of ways.
> >Q3
>
> Our primary historical comparison is to older models (GPT-3 and earlier), which prior work has shown to exhibit strong gender bias [3] (see L 80–89 & 244–246). While we also compare GPT-3.5 and GPT-4o-mini to show recent shifts, the main contrast is between current models and earlier generations. We’ll clarify this with an added sentence.
> >Q4
>
> Our findings align with prior work showing a shift toward female preference in GPT-3.5 and GPT-4o-mini. For example, [1] finds these models favor female candidates over male ones in hiring tasks, consistent with our observation that generated personas are more often women. However, other work shows stereotypical gender biases—e.g., GPT-3.5 assigns higher salaries to male candidates [2]. These results are not contradictory, as they involve different tasks, and increased female representation does not imply bias has been eliminated. Notably, [1,2] use gendered names in their prompts, while our analysis does not.
> >Q5
>
> Our gender association method is meant to be a lightweight component of our pipeline, and we evaluated its accuracy in Appendix A.1. We did not mean to claim that it is a significant technical innovation, and due to its simplicity, it is quite similar to many other gender inference methods – we will add text clarifying this in a revision.
> >Q6
>
> We appreciate the reviewer highlighting the importance of statistical significance. All words discussed in Section 5.3 are statistically significant identified with the Calibrated Marked Words method selected using p-values <0.05. Differences in SRBS between genders are also statistically significant, as confirmed by t-tests with p-values well below 0.05 across models (see Table 9 and Figure 2).
> However, we had not reported statistical significance for differences in Figure 3 between GPT-3.5 and GPT-4o-mini. We have now run a t-test on SRBS scores conditioned on gender and found the differences to be statistically significant. We will add this result to the appendix and note it in the main text.
> |Gender|Welch's t-statistic|p-value|
> |-|-|-|
> |F|2.20|0.03|
> |M|-2.16|0.04|
>
> We found the standard error for female representation in each occupation was consistently below 0.01 across all models, allowing us to estimate percentages to the nearest point with high confidence. For transparency, we will add a table of standard errors for each occupation-model pair to the appendix and note this in the main text.
>
> >Q7
>
> We do not discuss new stereotypes in our analysis, as we are not defending the validity of the stereotypes and harmful patterns surfaced, but rather are grounded in existing social science work (L 318-331) that point to the harm of these stereotypes and patterns.
>
> Misc improvement/Weakness 5: We thank the reviewer for identifying text improvements and will address in a revision.
>
> [1] Zhang et al. 2025. Hire Me or Not?...
>
> [2] Nghiem et al. 2024. "You Gotta be a Doctor, Lin": An Investigation...
>
> [3] Kirk et al. 2021. Bias out-of-the-box...

---

> > ### Comment · Reviewer_keQ9 · 2025-08-04
> >
> > Thanks, these rebuttals are reasonable and especially the proposed clarifications in various places will be helpful.  I am raising my score from 2 to 4.

---

> > > ### Author Response · Authors · 2025-08-09
> > >
> > > Thank you for raising your score from 2 to 4!! We greatly appreciate all of the points you brought up within your review and will be sure to include the proposed clarifications discussed in the rebuttal!!

---

### Official Review · Reviewer_Na2o · 2025-07-02

**Clarity:** 3
**Significance:** 3
**Originality:** 3
**Rating:** 5
**Confidence:** 3

**Summary:**

This paper investigates **persistent representational harms** in large language models (LLMs) despite increased demographic representation. Focusing on gender bias in occupational portrayals, the authors:
 (1) Analyze "who is represented: Generate biographies/personas for 63 occupations *without* specifying gender, finding women are significantly overrepresented (vs. U.S. BLS data) in GPT-3.5, GPT-4o-mini, and Llama-3.1-70b. Non-binary representation remains near zero.  (2) Develop novel methods:
 Gender Association Algorithm infers implicit gender in outputs via pronoun/honorific frequency (99.6% validation accuracy),
  Calibrated Marked Words identifies statistically significant words differentiating gendered descriptions (z-score ≥1.96), filtering common English terms and Subset Representational Bias Score quantifies bias by comparing word embeddings of *implicitly* vs. *explicitly* gendered outputs using Chamfer Distance.  (3) Examine "how people are represented": Reveal persistent stereotypes—e.g., women associated with *empathy*, *resilience*, and *passion*; men with *excellence* and *academia*—even in gender-neutral prompts. Biases intensify in newer models (e.g., GPT-4o-mini shows 100% increased bias for "software engineer").  (4) Critique bias mitigation: Increased female representation likely stems from interventions but fails to address *descriptive biases*, amplifying harmful narratives.

**Questions:**

Please see the weaknesses.

**Ethical Concerns:**

["NO or VERY MINOR ethics concerns only"]

**Final Justification:**

The author provided detailed rebuttals which are reasonable and especially the proposed clarifications in various places will be helpful. I am raising my score from 4 to 5.

**Limitations:**

Please see the weaknesses.

**Quality:**

3

**Strengths And Weaknesses:**

**Strengths**
1. **Novel methodological contributions**:
   - The **Gender Association Method** enables large-scale analysis of *implicit* gender biases in unmarked prompts, addressing a critical gap in bias evaluation.
   - **Subset Representational Bias Score** provides a quantifiable framework to compare biases across *associated* (implicit) vs. *specified* (explicit) gender contexts.
2. **Rigorous empirical design**:
   - Large-scale experiment: 63 occupations × 100+ generations/model (≈6K+ samples), validated across multiple SOTA LLMs.
   - Statistical robustness: *p*<0.05 for bias scores; careful calibration of marked words avoids noise from common terms (e.g., "the").
3. **Critical societal implications**:
   - Exposes limitations of current "diversity-focused" interventions (e.g., overrepresenting women without addressing stereotypes).
   - Highlights proliferation of neoliberal tropes (e.g., framing success as individual "resilience" vs. systemic change).
4. **Transparency**: Code/data released; detailed supplement (hyperparameters, validation, cluster analysis).

 **Weaknesses**
1. **Limited generalizability**:
   - U.S.-centric bias: Occupational gender distributions sourced from U.S. BLS, ignoring cultural variability.
   - Non-binary exclusion: <10% representation prevents meaningful analysis (acknowledged but unaddressed).
2. **Methodological constraints**:
   - **Pronoun dependency**: Gender Association Method may miss cultural/linguistic nuances (e.g., gender-neutral language).
   - **Subjective cluster interpretation**: Stereotype labels (e.g., "passion," "excellence") applied post hoc to word clusters lack inter-rater reliability metrics.
3. **Incomplete causal analysis**:
   - Speculates about "bias mitigation interventions" but lacks access to model training data/RLAIF details to verify causality.
   - No ablation on prompt phrasing impact (e.g., "biography" vs. "persona").
4. **Underdeveloped harm taxonomy**:
   - Groups all stereotypes under "representational harm" without severity gradation (e.g., "empathy" vs. "subservience").
   - Neglects intersectional biases (e.g., race/class + gender).

---

> ### Author Rebuttal · Authors · 2025-07-31
>
> We thank the reviewer for their review and appreciate that the reviewer found the work provides “novel methodological contributions,” has a “rigorous empirical design,” and discusses “critical social implications.”
>
> > U.S.-centric bias: Occupational gender distributions sourced from U.S. BLS, ignoring cultural variability.
>
> We do acknowledge this as a limitation in lines 372-374.
> > Non-binary exclusion: <10% representation prevents meaningful analysis (acknowledged but unaddressed).
>
> We do discuss non-binary representation in lines 208-210. Detailed figures are provided in Figure 6 in the appendix. Further analysis is minimal on non-binary representation due to the lack of text generated that can be associated with the non-binary gender identity.
> > Pronoun dependency: Gender Association Method may miss cultural/linguistic nuances (e.g., gender-neutral language).
>
> The Gender Association Method is tailored to our context, and although pronoun reliance is limiting, it is sufficient for our use case as demonstrated by the accuracy metrics of this method in Table 1 in the Appendix.
> |Gender| Correct % |Incorrect % |Not Captured %|
> |-|-|-|-|
> |Female |99.9180| 0.0080 |0.0740|
> |Male| 99.8463| 0.0053| 0.1483|
> |Non-binary |99.6693 |0.0037 |0.3280|
> > Subjective cluster interpretation: Stereotype labels (e.g., "passion," "excellence") applied post hoc to word clusters lack inter-rater reliability metrics.
>
> To mitigate the issue of subjective cluster labels, in a revision, we will replace the bar graphs with a table containing the actual clusters and their prevalence for each model and gender pair. An excerpt of the table is provided below:
> | Cluster                                 | GPT-3.5 %F | GPT-3.5 %M | GPT-3.5 # | GPT-4o-mini %F | GPT-4o-mini %M | GPT-4o-mini # | LLaMA-3.1 %F | LLaMA-3.1 %M | LLaMA-3.1 # |
> |-----------------------------------------|------------|------------|-----------|----------------|----------------|----------------|---------------|---------------|-------------|
> | empathy, empathize, empathetic          | 100.0      | 0.0        | 8         | 100.0          | 0.0            | 7              | 100.0         | 0.0           | 7           |
> | woman, actress, female                  | 100.0      | 0.0        | 7         | 100.0          | 0.0            | 17             | 90.91         | 9.09          | 11          |
> | shortterm, short                        | 100.0      | 0.0        | 4         | 25.0           | 75.0           | 4              | 20.0          | 80.0          | 5           |
> | advocate, advocates                     | 100.0      | 0.0        | 3         | 93.33          | 6.67           | 15             | 100.0         | 0.0           | 6           |
> | inspired, inspiration                   | 100.0      | 0.0        | 3         | 100.0          | 0.0            | 5              | 100.0         | 0.0           | 3           |
> | tireless, tirelessly                    | 100.0      | 0.0        | 3         | 100.0          | 0.0            | 5              | 83.33         | 16.67         | 6           |
> | decisions, decisionmaking, determination| 100.0      | 0.0        | 4         | 71.43          | 28.57          | 7              | 75.0          | 25.0          | 4           |
> | she, her, shes                          | 100.0      | 0.0        | 5         | 100.0          | 0.0            | 10             | 100.0         | 0.0           | 7           |
> | career, careers                         | 100.0      | 0.0        | 3         | 91.67          | 8.33           | 12             | 88.89         | 11.11         | 9           |
> | inclusive, inclusion, inclusivity       | 100.0      | 0.0        | 7         | 100.0          | 0.0            | 8              | 100.0         | 0.0           | 7           |
> | climbing, hiking, hiker                 | 100.0      | 0.0        | 5         | 75.0           | 25.0           | 4              | 75.0          | 25.0          | 8           |
> | prestigious                             | 100.0      | 0.0        | 3         | 80.0           | 20.0           | 5              | 100.0         | 0.0           | 3           |
> | practicing, training                    | 100.0      | 0.0        | 5         | 100.0          | 0.0            | 7              | 100.0         | 0.0           | 12          |
> | demands, demanding                      | 100.0      | 0.0        | 4         | 80.0           | 20.0           | 5              | 100.0         | 0.0           | 3           |
> | diversity, minorities, multicultural    | 100.0      | 0.0        | 5         | 100.0          | 0.0            | 6              | 100.0         | 0.0           | 7           |
> | herself                                 | 100.0      | 0.0        | 29        | 100.0          | 0.0            | 26             | 100.0         | 0.0           | 20          |
> | compassion, compassionate               | 100.0      | 0.0        | 10        | 100.0          | 0.0            | 12             | 100.0         | 0.0           | 10          |
> | yoga                                    | 100.0      | 0.0        | 5         | 100.0          | 0.0            | 5              | 100.0         | 0.0           | 5           |
> | passion, passions, passionate           | 90.0       | 10.0       | 10        | 100.0          | 0.0            | 8              | 100.0         | 0.0           | 6           |
> | families, familys, family               | 66.67      | 33.33      | 3         | 15.38          | 84.62          | 13             | 33.33         | 66.67         | 6           |
> | pursuits, pursuit, pursue, pursued, pursuing | 60.0  | 40.0       | 5         | 90.0           | 10.0           | 10             | 80.0          | 20.0          | 15          |
> | award, awardwinning, awards, accolades  | 60.0       | 40.0       | 5         | 75.0           | 25.0           | 4              | 85.71         | 14.29         | 7           |
> | inspiire, inspires, inspiring            | 60.0       | 40.0       | 5         | 87.5           | 12.5           | 8              | 100.0         | 0.0           | 4           |
> | countless, boundless                    | 33.33      | 66.67      | 3         | 100.0          | 0.0            | 3              | 75.0          | 25.0          | 4           |
> | husband, wife, spouse                   | 33.33      | 66.67      | 15        | 0.0            | 100.0          | 9              | 11.11         | 88.89         | 9           |
> | playing, gamer, gaming, games           | 0.0        | 100.0      | 8         | 0.0            | 100.0          | 7              | 0.0           | 100.0         | 7           |
> | basketball, sports                      | 0.0        | 100.0      | 4         | 0.0            | 100.0          | 10             | 0.0           | 100.0         | 10          |
> | tied, tie, ties                         | 0.0        | 100.0      | 3         | 100.0          | 0.0            | 3              | 60.0          | 40.0          | 5           |
> | his, himself, him                       | 0.0        | 100.0      | 29        | 0.0            | 100.0          | 28             | 0.0           | 100.0         | 27          |
> | charismatic                             | 0.0        | 100.0      | 3         | 100.0          | 0.0            | 4              | 0.0           | 100.0         | 5           |
> > Speculates about "bias mitigation interventions" but lacks access to model training data/RLAIF details to verify causality.
>
> As the reviewer pointed out, we “lack access to model training data/RLAIF details,” thus we are unable to explain why this shift in representation occurred. Thus, we join the call for greater transparency for model training, post-training methods, and bias-mitigation methods used to better understand and anticipate unintentional impacts of these methods.
> > No ablation on prompt phrasing impact (e.g., "biography" vs. "persona").
>
> We do provide gender distribution breakdowns by prompt type in Figure 5 of the Appendix.
> > Groups all stereotypes under "representational harm" without severity gradation (e.g., "empathy" vs. "subservience").
>
> Many representational harm taxonomies have been developed [1,2,3]. The purpose of our paper is not to develop another representational harm taxonomy, but rather, to demonstrate that some of the differences surfaced between generations associated between women and men are associated with representational harms and stereotypes. As it is outside the focus of our paper, we do not explicitly distinguish between severity of stereotypes and representational harms.
> > Neglects intersectional biases (e.g., race/class + gender).
>
> Our application of the evaluation methodology does not account for intersectional biases, but rather provides a detailed analysis for gender. We agree this is important work, and leave future work to extrapolate our evaluation methodology to intersectional groups and to develop a systematic contextually-aware algorithm for associating race/ethnicity, class, or other groups with generations.
>
> [1] Chien & David. 2024. Beyond Behaviorist Representational Harms: A Plan for Measurement and Mitigation
>
> [2] Katzman et al. 2023. Taxonomizing and measuring representational harms: A look at image tagging
>
> [3] Shelby et al. 2023. Sociotechnical harms of algorithmic systems: Scoping a taxonomy for harm reduction.

---

### Official Review · Reviewer_2KRr · 2025-07-02

**Clarity:** 3
**Significance:** 2
**Originality:** 2
**Rating:** 4
**Confidence:** 3

**Summary:**

This paper investigates a nuanced and critical dimension of representational harm in LLMs. The authors argue that merely increasing the representation of a demographic group (who is represented) does not automatically mitigate, and can even proliferate, stereotypical and harmful portrayals (how they are represented). To demonstrate this, the paper analyzes gender representation in occupational personas generated by state-of-the-art LLMs.

**Questions:**

* What do your results imply for practitioners? Would interventions at the RLHF or fine-tuning stages be more effective in targeting “how” people are represented?
* How is your Subset Representational Bias Score substantially different or more effective than other existing bias quantification tools, such as WEAT, StereoSet, or Marked Personas alone?
* Can your methods be applied to domains outside occupation modeling? For example, do you observe similar representational harms in education, family, or politics contexts?
* What about dealing with intersectional identities (e.g., race + gender)?
Misc:
* What does "English language" in line 153 refer to?
* What does "topic prior" in line 663 refer to? What dataset is it?

**Ethical Concerns:**

["NO or VERY MINOR ethics concerns only"]

**Final Justification:**

Sorry for the delay that I missed the final justification update in the first place. I originally misunderstood the novelty of the paper. After re-reading the paper, the rebuttal, as well as reviews from other reviewers, I have updated my score from 3 to 4.

**Limitations:**

yes

**Quality:**

3

**Strengths And Weaknesses:**

Strengths:
* The paper addresses a crucial and nuanced issue in AI safety.
* The proposed bias measurement pipeline is technically sound, particularly the enhancements to “Marked Personas.”

Weaknesses:
* Limited originality in framing: The underlying insight—that improving demographic parity (e.g., 50/50 gender distribution) doesn't eliminate harmful stereotypes—is widely discussed in fairness and safety literature.
* The study is narrowly focused on occupational prompts. It remains unclear how the findings extend to other real-world tasks (e.g., summarization, Q&A, dialogue).
* Details of the main methodology are pushed to the Appendix, making readers difficult to follow and understand the rigor of the proposed method.
* There is little discussion about how findings might guide practical LLM development beyond “do better.”
* Misc: clarify what A and B stand for in the Chamfer Distance formula.

---

> ### Author Rebuttal · Authors · 2025-07-31
>
> We thank the reviewer for their detailed review and insight, and especially appreciate how the reviewer pointed out that we “address a crucial and nuanced issue in AI safety” and our “proposed bias measurement pipeline is technically sound.”
> > What do your results imply for practitioners? Would interventions at the RLHF or fine-tuning stages be more effective in targeting “how” people are represented?
>
> Our results highlight the importance of addressing how groups are represented—not just who is represented, as we demonstrated how gender representational biases associated with how women and men are represented persist despite increased representation of women. Practitioners can use our evaluation methodology to detect representational biases and assess the impact of mitigation efforts on representational biases.
> While RLHF or fine-tuning may help reduce biases in how people are represented, further research is needed to evaluate how impactful these techniques would be, which we leave to future work. It is also essential to consider potential unintended effects of these interventions and utilize methods to surface or evaluate for these effects and implement means to best address these effects.
> >Can your methods be applied to domains outside occupation modeling? For example, do you observe similar representational harms in education, family, or politics contexts? What about dealing with intersectional identities (e.g., race + gender)?
>
> The general pipeline we propose applies to any situation where there is a protected group (e.g., women) and a non-protected group (e.g., men) in some context (e.g., generated biographies specific to an occupation), and we want to understand stereotypes specific to that context. In principle, the protected group could be selected based on gender, race, religion, politics, or any intersections thereof (provided it is sufficiently well-represented to obtain statistically meaningful conclusions), and the context does not need to be occupation-related. We chose to focus our paper on the specific protected group/context combination that we selected because we felt it was sufficiently demonstrative of our problem setting and methodology, and performing a thorough study of this particular case already merited a somewhat lengthy investigation. However, we believe it is a promising future direction to apply our techniques to other settings of interest, potentially augmenting our pipeline with more task-specific tools in case there are challenges that may arise outside of gender and biographies. To assist in utilizing our evaluation methodology in other contexts, the code we release will be modular and easy to use outside of the gender occupation domain of our empirical analysis.
>
> >How is your Subset Representational Bias Score substantially different or more effective than other existing bias quantification tools, such as WEAT, StereoSet, or Marked Personas alone?
>
> As described in Section 3.3, Chamfer-based SRBS solves the following problem: given a list of words C, and two lists of words T1 and T2, measure how much more similar C is to T1 than to T2. We note that in our setting, all of C, T1, and T2 have already been selected as statistically significant in their respective contexts (i.e., C is from generated personas with inferred genders, T1 is specified male, and T2 is specified female). We use the Chamfer distance to account for simple invariances in the word lists, e.g., synonyms should not be considered different.
>
> While WEAT has a similar goal to our SRBS evaluation method, it applies in a very different setting. WEAT modifies the human psychology Implicit Association Test (IAT) to evaluate biases in word embeddings. As such, it requires the lists of words to have particular characteristics that lend the test its meaning, making it much less flexible than our approach. For example, the gender-career IAT “categorizes typical male names and typical female names as well as words associated with career and family” [5]. In terms of the approach to measure the distance between words, WEAT relies on the mean cosine distance. The choice of mean of cosine distance is arguably appropriate for WEAT because the lists of attributes are carefully curated such that each list of attributes refers to a particular category (e.g., family). While we could indeed use mean cosine distance (and doing so would not affect the novelty of our proposed contribution), we believe Chamfer distance is much preferable because the interpretation and value of the mean cosine distance is less clear when the lists are not curated and may differ from one another in a multitude of ways.
>
> To give an illustrative example, the mean cosine distance used by WEAT applies best when the female word list has a coherent theme (e.g., {“she”, “mother”, “mrs.”}), and there exists a list of male words at the opposite extreme (e.g., {“he”, “father”, “mr.”}) However, in our setting, not only are the female stereotype words less coherent semantically (e.g., {“passionate”, “resilience”, “empathy”}) due to their context-specific usage, it is also unlikely that the antonyms of these words are representative of male biographies – just because women are described as passionate does not imply that men are described as dispassionate.
>
> On the other hand, StereoSet (more concretely, the CAT method in StereoSet) and Marked Personas are solving fundamentally different problems than the one addressed by our SRBS metric. CAT requires the specification of stereotype and anti-stereotype words (typically crowdsourced), and evaluates language models based on their preference of word. This is a different setting than we study; we do not provide any stereotype at all when prompting the language model, and automatically produce stereotypical word lists based on a statistical test (i.e., our Calibrated Marked Personas method, an earlier stage of our pipeline before applying the SRBS metric). Marked Words is a simpler variant of our Calibrated Marked Words method (which again, occurs at a different stage of our pipeline than the SRBS evaluation, and hence addresses a different problem of producing stereotypical word lists rather than evaluating their similarities); the difference lies entirely in the prior, see our response to Reviewer keQ9, Question 2 for details.
>
> > There is little discussion about how findings might guide practical LLM development beyond “do better.”
>
> We agree that diagnosing the root causes of these issues is challenging, in part due to limited transparency from model developers. Without access to training data, bias mitigation methods, fine-tuning practices, etc., it is difficult to understand which aspects related to model development have impacted model biases as well as how interactions between these aspects may have affected model biases. This makes providing more targeted recommendations to practitioners challenging. In response, our work offers a scalable evaluation methodology that practitioners can use to surface and track representational harms—an important step towards aiding practitioners in understanding whether representational biases persist after methods and techniques to address biases have been utilized. We also join ongoing calls for greater transparency to enable deeper diagnostic work [1].
> > Limited originality in framing: The underlying insight—that improving demographic parity (e.g., 50/50 gender distribution) doesn't eliminate harmful stereotypes—is widely discussed in fairness and safety literature.
>
> We agree that the limitations of demographic parity have been well-documented in the fairness and safety literature [2, 4]. However, to our knowledge, prior work has not examined how increased representation—particularly achieving or exceeding demographic parity—fails to mitigate how groups are represented in generative models. While some work has addressed the persistence of stereotypes in text-to-image models despite mitigation efforts [3], we are not aware of studies that demonstrate stereotype persistence under conditions of increased representation. Our findings show that for the majority of occupations analyzed, women are represented at or above 50%, yet gendered representational biases persist. If there is relevant literature we have missed, we would be grateful to be directed to it.
>
> > What does "English language" in line 153 refer to? What does "topic prior" in line 663 refer to? What dataset is it?
>
> The “English language” prior refers to the prior containing the distribution of words in the English language. The “topic prior” refers to the prior containing the distribution of words in the topic, which, in our case, is the generated biographies and personas. We will clarify this in Appendix A.2.
>
> [1] Cheng et al. 2023. Marked Personas: Using Natural Language Prompts to Measure Stereotypes in Language Models
>
> [2] Kenfack et al. 2024. A Survey on Fairness Without Demographics
>
> [3] Bianchi et al. 2023. Easily Accessible Text-to-Image Generation Amplifies Demographic Stereotypes at Large Scale
>
> [4] Mouzannar et al. 2020. From Fair Decision Making To Social Equality
>
> [5] Project Implicit Harvard

---

> ### Comment · Reviewer_2KRr · 2025-08-05
>
> Thank you for the clarification. I will maintain my score because I still have some concerns about the weaknesses mentionedabove.

---

> > ### Author Response · Authors · 2025-08-09
> >
> > Thank you for taking the time to read our rebuttal! Is there anything further we can clarify? Are there any weaknesses we have addressed (i.e. limited originality of framing, narrow focus on occupational prompts, etc.)? Furthermore, if the limited originality in framing is still an issue, do you mind pointing us to any literature that supports this framing? In our survey of the literature, we identified works discussing the limitations of demographic parity in algorithmic contexts for equality, but did not identify work discussing the limitation of demographic parity within the context of representation in generated text. Furthermore, our work demonstrates that increases in representation for women extend beyond demographic parity, and we demonstrate how this can lead to increased surfacing of representational harms. To our knowledge, previous work has not demonstrated this finding. If there is any literature we have missed, we would be grateful to be pointed towards it :)

---

### Official Review · Reviewer_JptD · 2025-07-08

**Clarity:** 4
**Significance:** 3
**Originality:** 3
**Rating:** 5
**Confidence:** 3

**Summary:**

The paper proposes that previous studies have failed to account for who and how gender biases were studied when it comes to representational harms. The authors propose a methodology that shows how word differences in describing occupations when it comes to gender distributions makes a difference in showing how the real representational harms are perpetuated

**Questions:**

Would be nice if the authors add more context in how such representational harms harm more women even when the occupations described here have more representation. That part was briefly mentioned in the conclusion but not given more context. Given the authors continued enagagement with this topic and how it adds an layer of insight to their input, it would be helpful to have this opinion or added references to this

**Ethical Concerns:**

["NO or VERY MINOR ethics concerns only"]

**Final Justification:**

I have raised the clarity score as I am satisfied with the rebuttal of authors to my review and the reviews of others. Rest of it I am keeping it as same.

**Quality:**

3

**Strengths And Weaknesses:**

It is an interesting point of research to understand word associations in gender based representional harms. This approach is much needed. To look into how words describing occupations vs how the words explicitly calling out genders in occupations is indeed very clever and very well done.

But overall the analysis and approach fails to dive deep into the question of generated image which in fact uses text to generate an image vs text to text generation. To quote the part here "although previous work has investigated gender occupational bias in generated94
images without specifying gender [ 36 ], to our knowledge, no previous work has investigated gender
occupational bias in generated text without specifying gender." Specifying or not specifying gender, image generating research in this area has indeed pointed this out many times.

---

> ### Author Rebuttal · Authors · 2025-07-31
>
> We thank the reviewer for their thoughtful engagement with our work and for recognizing the importance of studying who is represented within generative AI and how they are represented. We are especially grateful for the reviewer's appreciation of our methodology and the importance of this work.
>
> > overall the analysis and approach fails to dive deep into the question of generated image which in fact uses text to generate an image vs text to text generation
>
> We agree with the reviewer that previous work has investigated gender occupation bias in image generation. However, the task our paper focuses on is surfacing representational differences in text generation, which faces challenges distinct to those posed by images. In the context of images, researchers first associate an image with a gender using approaches such as gender classification models or using a VQA model to caption an image, as in [1]. With images, features related to the presentation of gender are prominent. Such features include hair length, facial features, makeup, and so on. In contrast, with generated biographies and personas, identifiers associated with gender are less prominent. In text, discussion of an individual’s appearance is often not included and primary gender markers are the pronouns used. Furthermore, studies on TTI, such as [1,2,3,4], focus on the distribution level representation but do not focus on differing features related to how groups are represented. Thus, although we agree work in image generation has demonstrated gender biases in occupation, our paper surfaces representational biases in how women and men are represented of which limited work exists within image generation, and we develop an evaluation methodology for surfacing these representational biases.
>
> > Would be nice if the authors add more context in how such representational harms harm more women even when the occupations described here have more representation.
>
> Representational harms may disproportionately affect women because they are now more frequently generated in model outputs. As representation increases, so does exposure to harmful stereotypes—e.g., if a biased portrayal appears in 90% of generated female biographies instead of 30%, the overall harm increases. Furthermore, the majority of stereotypes and harmful patterns analyzed related to women rather than men, as fewer statistically significant words for male generations were associated with stereotypes and representational harms. We'll expand upon these points within the paper to provide greater context.
>
> [1] Luccioni et al. 2023.Stable Bias: Evaluating Societal Representations in Diffusion Models
>
> [2] Naik & Nushi. 2023. Social Biases through the Text-to-Image Generation Lens
>
> [3] Cho et al. 2023. DALL-EVAL: Probing the Reasoning Skills and Social Biases of Text-to-Image Generation Models
>
> [4] Wan & Chang. 2025. The Male CEO and the Female Assistant: Evaluation and Mitigation of Gender Biases in Text-To-Image Generation of Dual Subjects

---

> > ### Comment · Reviewer_JptD · 2025-08-06
> > **Response to rebuttal**
> >
> > Thanks for your detailed answers to my review and also to other reviewers. As mentioned in my review, I think this is a much needed approach. I see that you have clarified a lot more clearly in your rebuttals, I only ask that such clarity is provided in the final camera ready version. I will consider raising my score.

---

> > > ### Author Response · Authors · 2025-08-09
> > >
> > > Thank you for considering raising your score!! We agree that this approach is much needed, and we will be sure to address all the changes discussed in the camera-ready version of the paper!

---

### Note · Authors · 2025-08-16

**To the AC(s)**: Thank you for your hard work in helping run NeurIPS.

We would like to bring to your attention a concern regarding the discussion with reviewer 2KRr. In their initial review, they noted “limited originality in framing” and suggested our key insights are “widely discussed in fairness and safety literature.” In our rebuttal, we explained why we believe our contributions are not present in existing work and highlighted how our methodology differs from related tools, which we feel are not well-suited for our setting. We also asked if they could point us to specific prior work matching their description.

In their follow-up, the reviewer maintained their original concerns but did not provide additional literature pointers or engage with our detailed clarifications. We recognize that reviewers may have time constraints, but we felt that this limited engagement made it difficult to address their concerns fully.

By contrast, reviewers JptD and keQ9 engaged in discussions, and either raised their score or expressed openness to doing so after our clarifications. While we agree that clarifying changes can be made to improve our presentation (and will make them in a revision, thanks to the helpful feedback we received), we remain concerned that our points on the value and originality of our work were not substantively considered by reviewer 2KRr.

We hope you will take this context into account when evaluating our submission. Thank you again for your efforts.

---

### Decision · Program_Chairs · 2025-09-17

**Decision:**

Accept (poster)

**Comment:**

My recommendation is to accept the paper.

The paper begins from the observation that, in occupational contexts, female representation has eclipsed male representation in some text generation tasks. However, the way that women are represented in these generation remains significantly different from men, and often aligns with stereotypes. The work proposes a new method for surfacing such stereotypes that can enable better diagnostics to address the _how_ representation problem, in addition to the _who_ representation problem.

Reviewers had some initial concerns about novelty and specifics of the proposal, but these concerns were largely resolved during the rebuttal period. The authors proposed some clarifications about the observation of representation percentage vs stereotyping, and the specifics of the methodology, including a schematic, that I hope they will follow through with in the camera ready.

One additional point: this work https://aclanthology.org/2025.acl-long.7/ makes a similar observation about representation in occupation personas, but doesn't make it a primary point. It may be worth citing.